# Numerical Investigation on Particle Erosion Characteristics of the Elbow Pipe in Gas-Steam Ejection Power System

**Qifei Chen** and **Guozhu Liang** *

School of Astronautics, Beihang University, Beijing 102206, China
* Correspondence: lgz@buaa.edu.cn

**Abstract:** In gas-steam ejection power systems, the $Al_2O_3$ particles in combustion products can cause severe erosion on the downstream elbow pipe. To calculate the particle erosion, a modelling approach is developed by combining a discrete phase model with a flow-thermal coupling model and introducing wall temperature parameters into the erosion model. Furthermore, the influence of particle size, total temperature and pressure, and particle mass flow rate was investigated. The results show that high temperature erosion depth can be expressed as the product of the time integral of temperature factor and the erosion rate at room temperature and is 1.63–3.57 times that at room temperature under different particle sizes. With the increase of particle size, the maximum erosion position tends to the inlet of the bend, and its value increases first and then decreases with the peak value 0.418 mm at particle diameter of 100 μm. The decrease in total temperature and total pressure reduces the erosion rate by reducing the particle velocity. The particle mass flow rate affects the gas-particle flow which, may cause the erosion to change greatly, especially when particle diameter is below 40 μm.

**Keywords:** gas-steam ejection power system; gas-particle flow; erosion; CFD; fluid-thermal coupling; particle size; particle mass flow rate

## 1. Introduction

Gas-steam ejection systems [1] are widely used in the launch of missiles for its simple structure, small size, safety, and reliability. The system uses a composite propellant gas generator containing aluminum powder, and its combustion products contain a large number of solid particles of $Al_2O_3$. These particles will impact and cause erosion on the inner wall of downstream components, in which the elbow pipe is most damaged by an abrupt change in the flow direction. The erosion of the elbow pipe can reduce its wall thickness and cause strength degradation, especially when the elbow pipe is used repeatedly. Thus, it is necessary to study the erosion characteristics of the elbow pipe in a gas-steam ejection system.

The phenomenon of particle erosion is widespread in the engineering field, and researchers initially mainly studied the erosion models of materials under different impact angles with different velocities, different kinds of particles and other parameters through the methods of mechanical theoretical analysis and experimental observation, and the classical ones are the Finnie model [2], Bitter model [3], and Levy model [4]. However, in engineering practice, the motion of particles itself is complex, and the erosion suffered by different devices in engineering cannot be obtained by material erosion models alone. Therefore, empirical erosion formulas based on a large number of experiments are proposed, which directly relate the erosion depth of the device to parameters such as inlet flow, velocity, and engineering device geometry, so as to easily and quickly calculate the erosion in engineering practice, such as Salama's formula in pipeline transmission, Bourgoyne's formula for steering gear erosion, etc. [5].

The formulation of empirical formulas for erosion requires a large number of experiments to be carried out, which is costly. Thanks to the development of numerical simulation methods for two-phase flow in recent decades, numerical methods have been developed in the field of erosion research. Through numerical simulation, the velocity of the particles on the wall, the angle of impact on the wall, and other parameters can be obtained, and the erosion caused by the particles can be substituted into the erosion model, and the sum of the erosion caused by different particles at the same location is the erosion suffered by the engineering device at that location during its work.

For the numerical simulation methods, since the 1990s, both Eulerian–Lagrangian [6–8] and Eulerian–Eulerian [9,10] methods have been applied to erosion calculations, but the Eulerian–Lagrangian method is more widely used because more easily facilitates obtaining the trajectories of particle motion. As the erosion calculation requires high accuracy of particle motion parameters, researchers have studied the near-wall grid size [11], the accuracy of the calculation of particles in the near-wall boundary layer [12], and the collision between particles in the two-phase flow simulation calculation [13,14]. These studies have developed to improve the numerical calculation methods of particle erosion.

With the gradual maturity of numerical calculation methods for particle erosion, many researchers have studied the problems in the erosion of engineering devices by using commercial calculation software. Peng [15] et al. compared the accuracy of erosion calculation of different erosion models under the same working conditions. Wang [16], Haide [17], and Hong [18] et al. studied the erosion characteristics of different devices for parameters such as inlet flow, particle size and pipe diameter.

However, the above studies mainly focus on normal temperature erosion, and relatively little research has been performed on particle erosion accompanied by high-temperature two-phase flow heat transfer. Particle erosion with high-temperature two-phase flow heat transfer is widely seen in structures involving metallized solid propellants, such as particle-to-throat lining in solid rocket motors [19–22], applied to the inner wall of launch tubes [23], and to the elbow of the ejection power system studied in this paper. In these studies, heat transfer was not considered, or wall temperature was not introduced into the particle erosion model. Thus, a modeling approach that considers the coupling of erosion and heat transfer needs to be developed.

In this paper, a modelling approach is developed by combining a discrete phase model (DPM) with a flow-thermal coupling model and introducing wall temperature parameters into the erosion model. After verification by the experimental data on erosion at room temperature and heat transfer in high temperature gas-particle flow, the approach is applied to calculate the particle erosion of the elbow pipe in gas-steam ejection power systems in high temperature gas-particle flow. Furthermore, the influence of particle size, total inlet temperature and pressure, and mass flow rate of particle on elbow pipe erosion is also investigated.

## 2. Engineering Problem Description

### 2.1. Structure and Working Condition

A schematic diagram of the elbow pipe is shown in Figure 1. The upstream combustion gas-steam mixture and $Al_2O_3$ particles enter from the inlet of the bottom head and then flow out from the outlet of the elbow pipe. The total temperature of flow at the inlet is about 1473 K, and the total pressure is about 4 MPa. The gas pressure at the elbow pipe outlet is one atmosphere. Before operation, the structure is at room temperature. The aluminum content of hydroxyl terminated polybutadiene propellant (HTPB) is about 6%. Thus, the mass of $Al_2O_3$ particles in combustion products is about 12% and the particle mass flow rate is estimated to be 25 kg/s. The flow lasts for approximately 0.7 s.

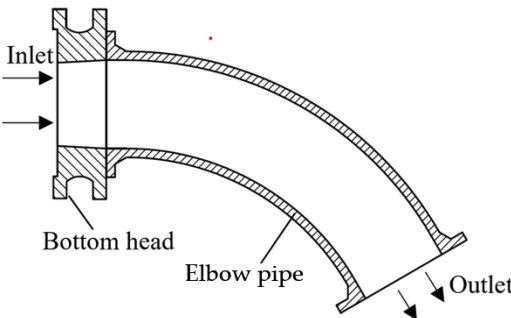

**Figure 1.** Elbow pipe structure of an ejection power system.

*2.2. Material Parameter*

According to the thermodynamic calculation of the HTPB, the gas transport properties at different temperatures can be obtained. After polynomial fitting, the analytic expression of dynamic viscosity, thermal conductivity, and specific heat capacity of gas are

$$\eta_g = -2.439 \times 10^{-12}T^2 + 3.112 \times 10^{-8}T \tag{1}$$

$$\lambda_g = -9.038 \times 10^{-9}T^2 + 1.154 \times 10^{-4}T \tag{2}$$

$$c_{p,g} = -4.571 \times 10^{-5}T^2 + 0.3137 \tag{3}$$

According to the International Association for the Properties of Water and Steam [24] (IAPWS-97), the analytic expression of the dynamic viscosity, thermal conductivity, and specific heat capacity of steam are fitted as

$$\eta_{st} = 3.928 \times 10^{-8}T + 1.778 \times 10^{-6} \tag{4}$$

$$\lambda_{st} = 1.331 \times 10^{-4}T - 0.0359 \tag{5}$$

$$c_{p,st} = 0.6927T + 1599 \tag{6}$$

In Formulas (1)–(6), the units of $T$, $\eta$, $\lambda$, $c_p$ are K, Pa·s., W·m$^{-1}$K$^{-1}$, and J·kg$^{-1}$K$^{-1}$ respectively. The subscript g and st indicates gas and steam, respectively.

The chemical composition and thermo-physical properties of the elbow pipe material, boiler steel 12Cr1MoV, are shown in Tables 1 and 2.

**Table 1.** Chemical composition of 12Cr1MoV steel.

| C | Si | Mn | Cr | Mo | V | P | S | Cu |
|---|---|---|---|---|---|---|---|---|
| 0.12 | 0.17–0.37 | 0.4–0.7 | 1 | 0.25–0.35 | 0.15–0.3 | <0.35 | <0.35 | <0.25 |

**Table 2.** Thermo-physical properties of 12Cr1MoV steel.

| Materials | $\lambda_s$/W·m$^{-1}$·K$^{-1}$ | | | | $c_s$/J·kg$^{-1}$·K$^{-1}$ | | | | $\rho_s$/kg·m$^{-3}$ |
|---|---|---|---|---|---|---|---|---|---|
| 12Cr1MoV | 293K | 473K | 673K | 873K | 293K | 473K | 673K | 873K | 7860 |
| | 45.2 | 45.2 | 40.5 | 35.5 | 560 | 586 | 653 | 729 | |

## 3. Physical and Mathematical Modeling

*3.1. Physical Model*

The key to the above engineering problem is the particle erosion in high temperature gas-particle flows, for which physical modeling is carried out. The process of particle erosion is shown in the Figure 2. A particle impacts a wall with velocity $v_{pi}$ and impact angle $\theta$ and bounces off with velocity $v_{pe}$, causing the removal of wall material. Heat transfer also occurs between the two-phase flow and the wall. Along with the heat transfer

$Q_{gw}$ between the gas and the wall, the particle also influences the heat transfer process in two other ways. The first is direct heat transfer $Q_{pw}$ between the particle and the wall, while the second is indirect heat transfer $Q_{pg}$ through the gas surrounding the particle. As a result of the heat transfer, the wall temperature increases, which affects particle erosion on the wall significantly.

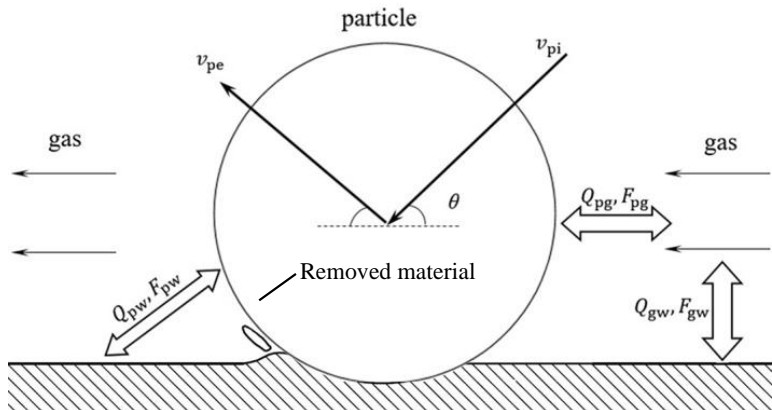

**Figure 2.** Diagram of erosion and heat transfer in high temperature gas-particle flow.

In order to calculate the erosion in high temperature gas-particle flow, the following assumptions are made.

1. In the flow especially near the wall, the effect of the volume of the particles on the flow and heat transfer can be neglected. As a result, the discrete-phase approach that ignores the volume of particles can be used in the flow near the wall.

2. The heat transfer between the particles and the wall can be neglected compared to the particles passing through the surrounding gas to the wall.

3. The effect of the change in wall morphology caused by the erosion on heat transfer can be neglected, i.e., it is considered that only the one-way coupling of heat transfer to erosion is considered in the calculation of high temperature gas-particle two-phase flow to wall erosion.

The mathematical models for gas phase, particle phase, structural heat transfer, and particle erosion are presented below.

### 3.2. Mathematical Model of Gas Phase

The gas flow in the elbow pipe is regarded as frozen flow, without considering chemical reaction and ignoring the viscous dissipation energy term. Then, the conservation equations of continuity, momentum, and energy are

$$\frac{\partial \rho_f}{\partial t} + \frac{\partial (\rho_f u_j)}{\partial x_j} = 0 \tag{7}$$

$$\frac{\partial (\rho_f u_i)}{\partial t} + \frac{\partial (\rho_f u_i u_j)}{\partial x_j} = -\frac{\partial p}{\partial x_i} + \frac{\partial}{\partial x_j}\left( (\mu + \mu_t)\left( \frac{\partial u_i}{\partial x_j} + \frac{\partial u_j}{\partial x_i} \right) \right) + S_{F,i} \tag{8}$$

$$\frac{\partial (\rho_f h_{tot})}{\partial t} - \frac{\partial p}{\partial t} + \frac{\partial (\rho_f u_j h_{tot})}{\partial x_j} = \frac{\partial}{\partial x_j}\left( \lambda \frac{\partial T}{\partial x_j} + \frac{\mu_t}{Pr_t} \frac{\partial h}{\partial x_j} \right) + S_Q \tag{9}$$

where $\rho_f$, $u$, $p$, $\mu$, $\mu_t$, $h_{tot}$, $h$, and $\lambda$ represent the density, velocity, pressure, viscosity, turbulence viscosity, total enthalpy, static enthalpy, and heat conductivity of the gas respectively. $Pr_t$ is the turbulent Prandtl number. $S_{F,i}$, $S_Q$ are the momentum source and energy source respectively, which are transferred from particles to the gas.

Moreover, the conservation equations need to be supplemented by the gas equation of state. The combustion gas is generally regarded as an ideal gas, but the water vapor is

regarded as real gas. According to the basic equations in the 2nd and 5th zones given by IAPWS-97 [24], the state equation of the water vapor gas at 700–1400 K is as follows

$$pv = RT\left[1 + \pi\left(0.1058\tau^3 - 0.3448\tau^2 + 0.383\tau - 0.1544\right)\right] \tag{10}$$

where $\pi = \frac{p}{p_r}$, $p_r = 1$ MPa, $\tau = \frac{T}{T_r}$, $T_r = 1000$ K, and $R = 461.9$ J/(kg·K).

### 3.3. Mathematical Model of Particle Model

The momentum equation for translation of a particle can be described by Newton's kinetic equation

$$m_p\frac{du_p}{dt} = F_D \tag{11}$$

where $m_p$, $u_p$, and $F_D$ are the mass of a single particle, particle velocity, and the drag force on the particle by the gas, respectively.

$$F_D = \frac{1}{2}C_D\rho_f A_p|u_f - u_p|(u_f - u_p) \tag{12}$$

where $\rho_f$, $u_f$, and $A_p$ are the fluid density, fluid velocity, and the effective particle cross section, respectively. $C_D$ is the drag coefficient, evaluated by Schiller and Naumann [25] as

$$\begin{cases} C_D = 24Re_p^{-1}\left(1 + 0.15Re_p^{0.687}\right) & \text{if } Re_p < 1000 \\ C_D = 0.44 & \text{if } Re_p > 1000 \end{cases} \tag{13}$$

where $Re_p$ is the particle Reynolds number.

When particles collide with the wall, a restitution model is needed to calculate the velocities of the particle after the collision. The restitution model proposed by Grant and Tabakoff [26] is as follows

$$e_{per} = 0.993 - 1.76\theta + 1.56\theta^2 - 0.49\theta^3 \tag{14}$$

$$e_{par} = 0.998 - 1.55\theta + 2.11\theta^2 - 0.67\theta^3 \tag{15}$$

where $e_{per}$ and $e_{par}$ are the ratio of velocity after and before the collision, in perpendicular and parallel directions respectively.

The temperature of a particle satisfies the energy conservation equation

$$m_p c_p \frac{dT_p}{dt} = Q_C \tag{16}$$

where $c_p$ and $T_p$ are the specific heat and temperature of the particle, respectively. $Q_C$ is the heat exchange of gas to the particle, which can be calculated by

$$Q_C = \pi d\lambda_f Nu\left(T_f - T_p\right) \tag{17}$$

The empirical formula of $Nu$ was proposed by Ranz Marshall [27] as

$$Nu = 2 + 0.6Re^{0.5}\left(\mu\frac{c_{f,p}}{\lambda_f}\right)^{\frac{1}{3}} \tag{18}$$

where $c_{f,p}$, $\lambda_f$, and $\mu$ are specific heat at constant pressure, thermal conductivity, and viscosity coefficient of fluid, respectively.

In gas-particle flow, Li et al. [28] argued that the direct heat transfer of particle-wall collision can be ignored when the solid particles concentration is low, and Rousta et al. [29] also found that the direct heat transfer is significantly small compared to the indirect heat

transfer of particle-gas-wall. Therefore, the heat transfer when the particles collide with the wall is not considered in this paper.

### 3.4. Mathematical Model of Solid Heat Transfer

The heat conduction equation of the solid is

$$\frac{\partial(T_s)}{\partial t} = \nabla \cdot \left( \frac{\lambda_s}{\rho_s c_s} \nabla T_s \right) \tag{19}$$

where $\rho_s$, $\lambda_s$, $c_s$, and $T_s$ are the density, thermal conductivity, specific heat, and temperature of the solid material, respectively.

The continuity conditions should be satisfied for temperature and heat flux at the fluid–solid interface, namely

$$T_{f,w} = T_{s,w} \ , \ \lambda_f \left.\frac{\partial T_f}{\partial n_f}\right|_w = \lambda_s \left.\frac{\partial T_s}{\partial n_s}\right|_w \tag{20}$$

where $n$ represents the normal direction of the interface, and subscript w means the parameters at the wall.

### 3.5. Mathematical Model of Particle Erosion

The erosion is measured by the penetration rate $\dot{E}$ and depth $E$ per unit area, which can be calculated, respectively, as

$$\dot{E} = \frac{1}{A\rho_{elb}} \sum_{i(A)} \dot{m}_i e_{r,i} \tag{21}$$

$$E(t) = \int_0^t \dot{E} \mathrm{d}t \tag{22}$$

where $\rho_{elb}$ is the density of elbow pipe material, $A$ is the impact face area and equal to the grid cell area in numerical calculations, $\dot{m}_i$ is the mass of each particle that collides with the face in unit time, and $e_{r,i}$ is the erosion ratio of the particle.

In addition to the materials of particle and target, the erosion ratio of metal material by solid particles is related to the impact velocity, impact angle, particle size, temperature, etc. The general formula is

$$e_r = Kg(\theta)\left(\frac{v}{v_r}\right)^n f(d_p)h(T_w) \tag{23}$$

where $K$ is erosion constant, $g(\theta)$ is a function of impact angle $\theta$, $v$ is impact velocity, $v_r$ is the reference temperature, $n$ is velocity exponent, $f(d_p)$ is a function of particle size $d_p$, and $h(T_w)$ is a function of impact wall temperature $T_w$.

The elbow pipe in the gas-steam ejection power system is made of boiler steel 12Cr1MoV, which is suitable for high temperatures, whereas the widely used E/CRC erosion model [30], Oka's erosion model [31,32] do not take temperature into account. Das [33] et al. conducted erosion studies of boiler steel 1.25Cr-1Mo-V with 100 μm $Al_2O_3$ particles at different impact velocities, angles, and temperatures. Due to the close chemical composition and mechanical properties of the two steels, the study of Das can be used to fit the erosion model for the elbow pipe erosion in this paper.

According to the experimental results by Das [33], the correlations are

$$g(\theta) = 5.703 \times 10^{-9}\theta^5 - 1.519 \times 10^{-6}\theta^4 + 1.537 \times 10^{-4}\theta^3 - 7.49 \times 10^{-3}\theta^2 + 0.1735\theta \tag{24}$$

$$h(T_w) = 0.003798\left(\frac{T_w}{T_r}\right)^2 - 0.01270\left(\frac{T_w}{T_r}\right) + 1.009 \tag{25}$$

$$n = 2.28 \tag{26}$$

$$T_\text{r} = 298 \text{ K}, v_\text{r} = 150 \text{ m/s} \tag{27}$$

where $T_\text{r}$ is the reference temperature.

Based on extensive particle erosion experiments, Uzi et al. [34] put forward the general formula of the particle size function as

$$f\left(d_\text{p}\right) = 1 - e^{\left(-\frac{3(d_\text{p} - d_\text{min})}{d_\text{c}}\right)} \tag{28}$$

where $d_\text{c}$ is the critical particle size and $d_\text{min}$ is the threshold particle size. It is also proposed by Uzi [34] that the critical particle size of $Al_2O_3$ is 200 μm and the threshold particle size is 1–5 μm. In the present research, 1 μm is taken as the threshold particle size considering the high impact velocity and high temperature.

$$d_\text{c} = 200 \text{ μm}, \quad d_\text{min} = 1 \text{ μm} \tag{29}$$

Finally,

$$K = 23.14 \tag{30}$$

## 4. Numerical Method and Validation

### 4.1. Numerical Method

The numerical method to calculate the particle erosion in high temperature gas-particle flow is indicated in Figure 3. The discrete phase model with two-way coupling and the fluid-thermal coupling model are combined by the fluid control equations to calculate the velocity and angle of the particles hitting the surface of the structure, as well as the temperature of the surface of the structure, which can be imported into the particle erosion model to calculate the erosion value.

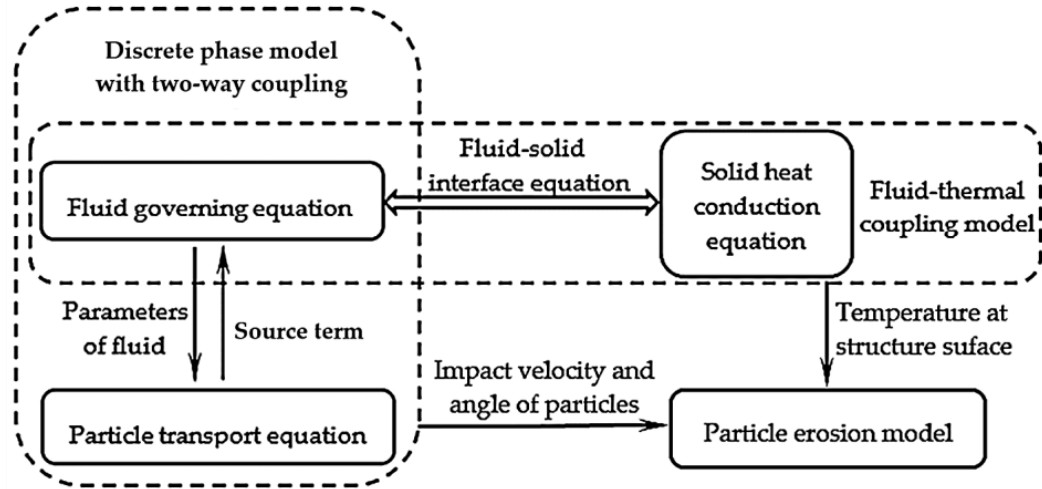

**Figure 3.** Numerical model coupling the gas-particle flow and the structure temperature.

The commercial software ANSYS CFX is adopted to conduct the numerical simulations. In ANSYS CFX, for the DPM, at the location of the particle, the fluid transfers its parameters, such as velocity, temperature, density, etc., to the particle transport equation, while the particle inputs the particle-to-fluid forces and heat transfer into the fluid control equation in the form of a source term. For the fluid-thermal coupling model, CFX solves the gas energy equation, the solid thermal conductivity equation, and the fluid-solid interface equation jointly after coupled solution of the continuity equation and momentum equation, in every iteration. After solving the above equations, the velocity and angle of the particles impacting the wall, as well as the temperature of the surface of the structure, are imported

into the Fortran subroutine written for the particle erosion model, through the User Fortran Routine interface in ANSYS CFX.

The mesh model of the flow field and structure of the elbow is shown in Figure 4. The red grid and blue grid represent the flow field mesh and structure mesh, respectively. The nodes of the fluid-solid interface correspond one by one. The flow field mesh has a radial size of 5 mm and an axial size of 10 mm. The mesh of flow field near the wall is gradually refined, where the first cell height of the boundary layer grid inside and outside the elbow pipe are 0.00045 mm and 0. 00015 mm respectively, and the increasing ratio of the both are 1.2. The y+ of boundary layer grids are all near 1, which meet the requirement of adopting the low Reynolds number model for the wall for use with the SST turbulence model.

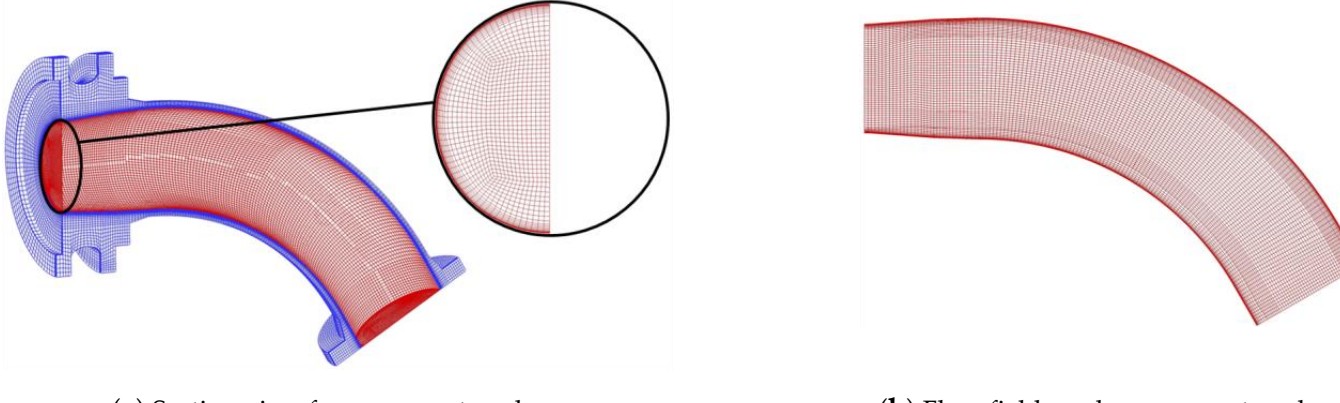

(**a**) Section view from symmetry plane        (**b**) Flow field mesh on symmetry plane

**Figure 4.** Mesh model of the flow field and structural of elbow pipe.

The main concern is the structure temperature near the fluid–solid interface, where there is strong nonlinearity in transient heat transfer problems. Thus, the mesh size in the solid region at the interface should be verified for mesh independence. The sizes and results are shown in Table 3, where the calculation time is 0.1 s.

**Table 3.** Inner wall surface mesh size independence.

| Mesh Size at Fluid-Solid Interface/mm | Wall Temperature/K |
| --- | --- |
| 0.18 | 522.3 |
| 0.36 | 520.7 |
| 0.09 | 522.9 |

As can be seen from Table 2, when the boundary mesh size of the inner wall of the structure is taken as 0.18 mm, the calculated temperature variation with size is already less than 1%, which satisfies the mesh irrelevance test, then the solid wall mesh size is divided into 0.18 mm. The entire mesh model consists of 1,574,778 elements.

The SST model [35], the second order upwind scheme, and the second order backward Eulerian scheme are adopted for the turbulence model, the advection term, and the time advance, respectively.

The lower head inlet is given a total temperature of 1473 K and a total pressure of 4 MPa, and the exit of the bend is given a gas pressure of one atmosphere. The entire power bend structure is set to adiabatic wall conditions on the outer wall surface, and the thermal resistance between components is set to 0. The mass flow rate of particles is 25 kg/s, and the number of particles in the simulation is $2,000,000$ s$^{-1}$. The particle diameter is set to a uniform 100 μm at first. The total time is 0.7 s, and the time step is 0.0001 s.

### 4.2. Validation

Due to the lack of experiments on particle erosion in high-temperature gas-particle flows, this method will be validated in two parts, namely the particle erosion calculation in elbow at room temperature and the heat transfer calculation in high temperature gas-particle flow.

#### 4.2.1. Validation for Particle Erosion at Room Temperature

An experiment on the erosion of an elbow by air-sand flow is employed in this work to verify the calculation method concerning erosion [15]. The geometry model and mesh model of elbow pipe flow field are shown in Figure 5. The diameter and curvature radius are 41 mm and 133.25 mm respectively. The hexahedral structured mesh is adopted to mesh the whole volume, and the grid size is around 2 mm. The grid in the near-wall region is gradually refined and the bottom grid size is 0.014 mm so that the y+ is around 1, which can lead to higher accuracy of the erosion calculation, according to the study by Karimi et al. The grid number used in this case is approximately 831,618.

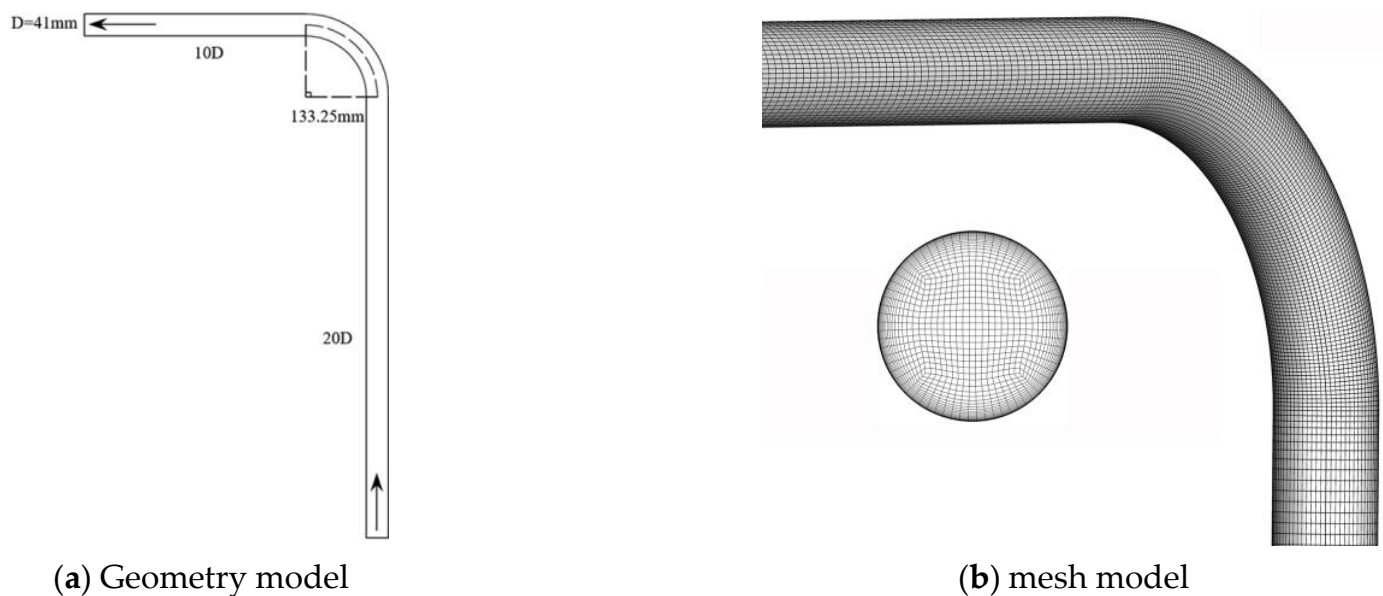

(**a**) Geometry model          (**b**) mesh model

**Figure 5.** Geometry model and mesh model of elbow pipe flow field.

The E/CRC erosion model is adopted in the calculation and the equations are

$$e_r = C(BH)^{-0.59}F_s v_p^n g(\alpha) \tag{31}$$

$$g(\alpha) = \sum_{i=1}^{5} R_i \alpha^i \tag{32}$$

where $BH$ and $F_s$ are the Brinell hardness of the target material and the particle sharpness factor, equaling to 120 and 0.2 respectively. $C$, $n$, and $R_i$ are empirical constant and can be found in reference [15].

Figure 6 shows the comparison of the calculated and the experimental erosion rate along elbow curvature angle. From Figure 6, it can be seen that the experimental data and the calculated results agree well at all points except 50° and 70°. For the fluctuation of solid grain erosion itself, it can be considered that the erosion calculation method in this paper is suitable.

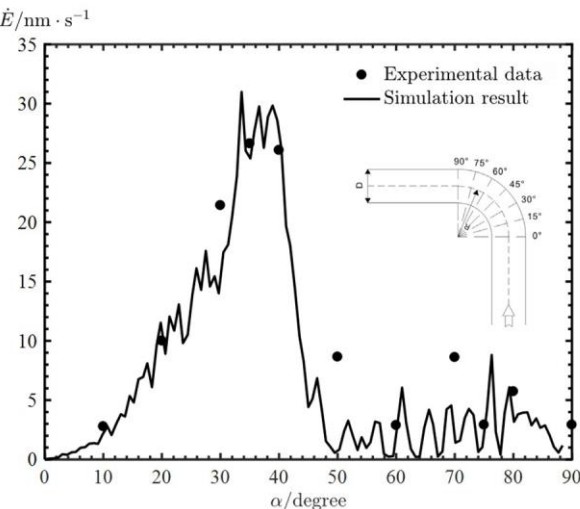

**Figure 6.** Comparison of the experimental data and simulation result on erosion rate.

### 4.2.2. Validation for Heat Transfer in High Temperature Gas-Particle Flow

To verify the calculation of heat transfer under high temperature gas-particle flow, the experimental data of the nozzle in the solid rocket motor (SRM) by Liu [36] are adopted. The geometry model and mesh models of the nozzle and its flow field are shown in Figure 7. The nozzle constriction and expansion angles are 90 and 10 respectively, with a throat diameter of 11.4 mm. The inner layer of the nozzle expansion is graphite with a thickness of 12 mm. The temperature measurement points are located in the red area, where points A and B are both 8 mm deep in the graphite layer and axially located at expansion ratios of 2.38 and 3.21, respectively. The mesh models of nozzle and its flow field are somewhat simplified compared to the engineering model, but the key dimensions remain the same. The hexahedral structured mesh is adopted to mesh the whole volume, and the grid size is from 1 to 3 mm. The near-wall region is gradually refined, and the bottom grid size is 0.014 mm so that the y+ is approximately 1, which can meet the SST turbulence model for heat transfer calculation. The grid number used in this case is approximately 195,776.

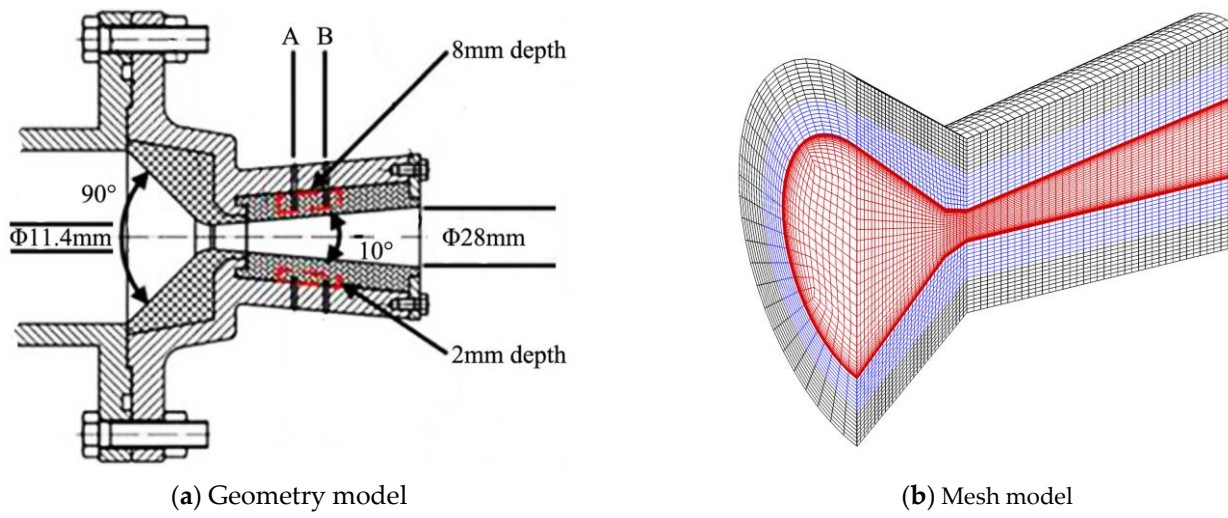

(**a**) Geometry model                 (**b**) Mesh model

**Figure 7.** Geometry model and mesh model of the SRM.

This SRM uses a butyl hydroxyl propellant containing 17% Al and its gas and particle parameters as well as inlet boundary conditions are set with reference to the literature. The time step for the transient calculation was 0.00002 s and the calculation duration was the first 1.6 s after the SRM was started. The calculated results of temperature at point A and B

are compared with the experimental data of as shown in Figure 8. As can be seen from the Figure 5, except the value at 0.4 s, the deviations between the calculated and experimental data are within approximately 5%, which is acceptable for the numerical calculations.

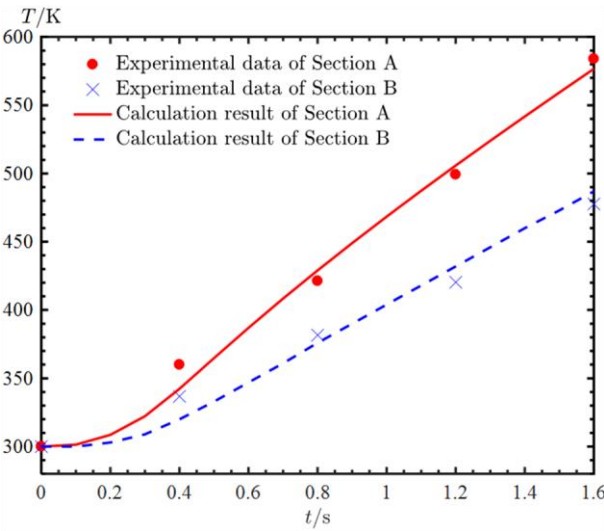

**Figure 8.** Comparison of the experimental data and calculation results on nozzle temperature.

## 5. Result and Discussion

### 5.1. Influence of Temperature on Transient Erosion

Under the above working conditions, the calculation results of the particle trajectories and the penetration depth of erosion at 0.7 s are shown in Figure 4. As can be seen in Figure 9a, the particles flow straight in from the inlet, colliding with the elbow pipe wall and deflecting to the other side. Deflecting particles cause the particle distribution area to take on a V-shape, which is similar to the results in other studies [37–39]. In Figure 9b, the distribution of penetration depth shows the same shape, and the maximum value is located in the cross-center region. The maximum penetration depth of erosion is calculated as 0.418 mm.

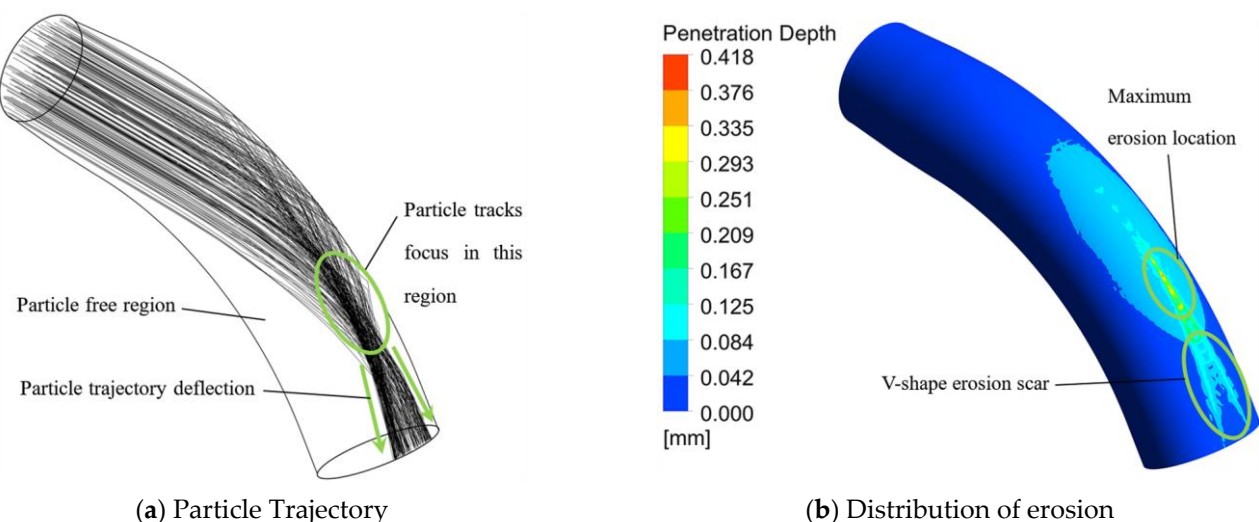

(**a**) Particle Trajectory                    (**b**) Distribution of erosion

**Figure 9.** Particle trajectory and erosion distribution in the elbow pipe.

In order to investigate how temperature affects transient erosion, Figure 10 illustrates penetration depth and temperature variation with time at the location of maximum penetration depth, which shows that erosion is not linearly related to time.

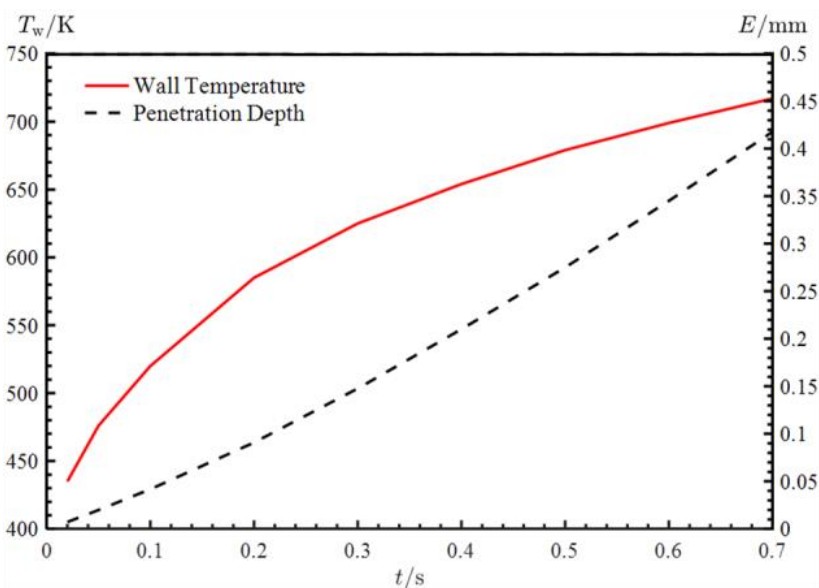

**Figure 10.** Time curves of the penetration depth and wall temperature.

It may be assumed that this nonlinearity is entirely due to the variation of temperature, and then it can be derived from the erosion equation as follows

$$e_r = Kg(\theta)\left(\frac{v}{v_r}\right)^n f(d_p)h(T_w) = e_{r,s}h(T_w) \tag{33}$$

$$\dot{E} = \frac{h(T_w)}{A\rho_{elb}}\sum_{i(A)} \dot{m}_i e_{r,s,i} = \frac{h(T_w)}{h(T_{ref})}\frac{h(T_{ref})}{A\rho_{elb}}\sum_{i(A)} \dot{m}_i e_{r,s,i} = \frac{h(T_w)}{h(T_{ref})}\dot{E}_s \tag{34}$$

$$E(t) = \int_0^t \dot{E}dt = \dot{E}_s \int_0^t \frac{h(T_w)}{h(T_{ref})}dt = \dot{E}_s H(t)\cdot t \tag{35}$$

where $\dot{E}_s$ represents the steady-state erosion rate at room temperature, and $H(t)$ represents the multiplier of particle erosion due to heat transfer in the high temperature gas-particle flow at moment $t$. Define $H_T = H(t_f)$, representing the multiple at the final time, where $t_f$ is the final time.

To verify the hypothesis, using the simulation results $E(t)$ and $T_w(t)$, let

$$\dot{E}'_s(t) = \frac{E(t)}{H(t)} \tag{36}$$

and

$$E'(t) = \dot{E}'_s\big|_{t=0.1s}\cdot H(t) \tag{37}$$

The hypothesis holds if $\dot{E}'_s(t)$ varies little with time, as well as $E'(t)$ are very close to $E(t)$.

The results are shown in Figure 11. The $\dot{E}'_s(t)$ remains constant except for the first 0.02 s when particles just start to flow into the elbow pipe, and $E'(t)$ is very close to $E(t)$. As a result, it can be concluded that only the temperature factor affects the erosion rate in transient erosion of the elbow pipe.

### 5.2. Influence of Particle Size on Erosion

Liu [40] and Li et al. [41] propose that the particle diameter in solid rocket motors range from a few microns to several hundred microns. To study the effect of particle size on erosion, the erosions of particles in the diameter range of 8–300 μm were simulated. The simulation results of solid particle distribution and the penetration depth of erosion by 8 μm, 20 μm, 40 μm, and 300 μm particles are shown in Figure 12, respectively. Compared

with Figure 9, the small particles do not cross the symmetry plane of the elbow pipe after bouncing off the inner wall, thus there is no bifurcation in the erosion distribution of the inner wall of the elbow pipe. According to the drag force equation, the acceleration of the particle by the drag force is inversely proportional to the particle size, and the small particles are more likely to flow together with the gas phase, so that on the one hand, the particle trajectory after bouncing has a small angle with the elbow pipe symmetry surface, and on the other hand, it is more likely to be blown to the inner wall by the gas phase so that the bounce will hit the same side of the inner wall of the elbow pipe. As the particle size increases, the acceleration of the particle by the gas phase traction decreases, and the particle will cause erosion on the other side of the inner wall of the elbow pipe after bouncing, thus causing the bifurcation of the erosion distribution.

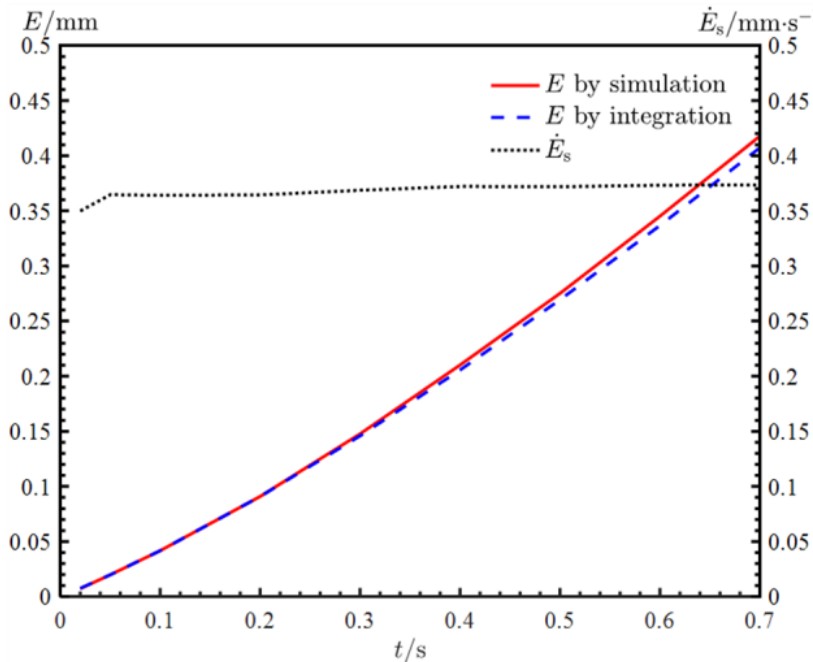

**Figure 11.** $\dot{E}_s$ and the comparison of the simulation result and integration result of $E$.

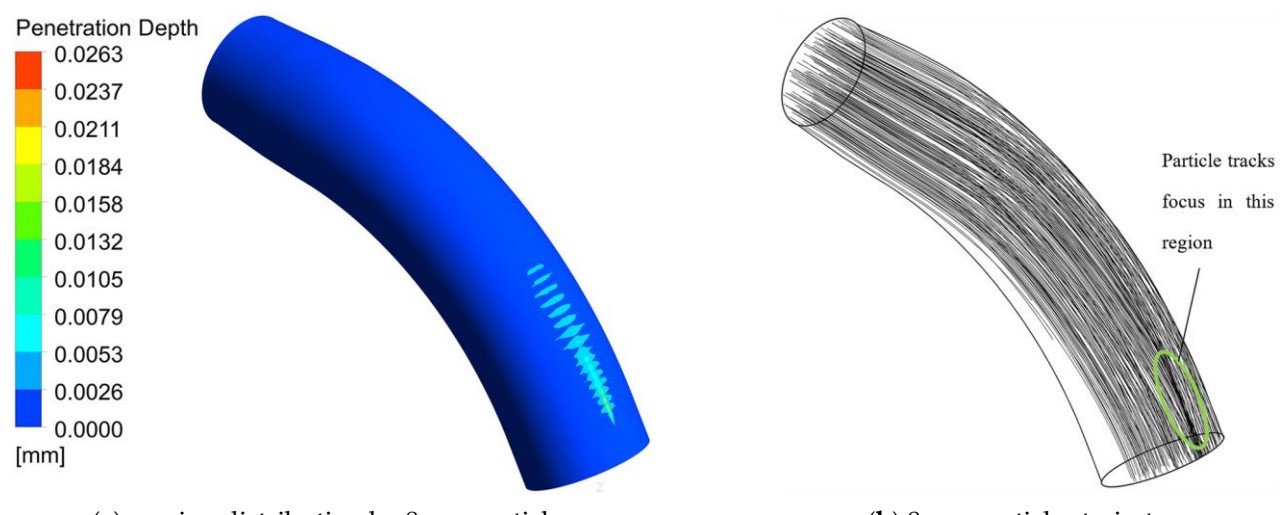

(**a**) erosion distribution by 8 μm particles

(**b**) 8 μm particles trajectory

**Figure 12.** *Cont.*

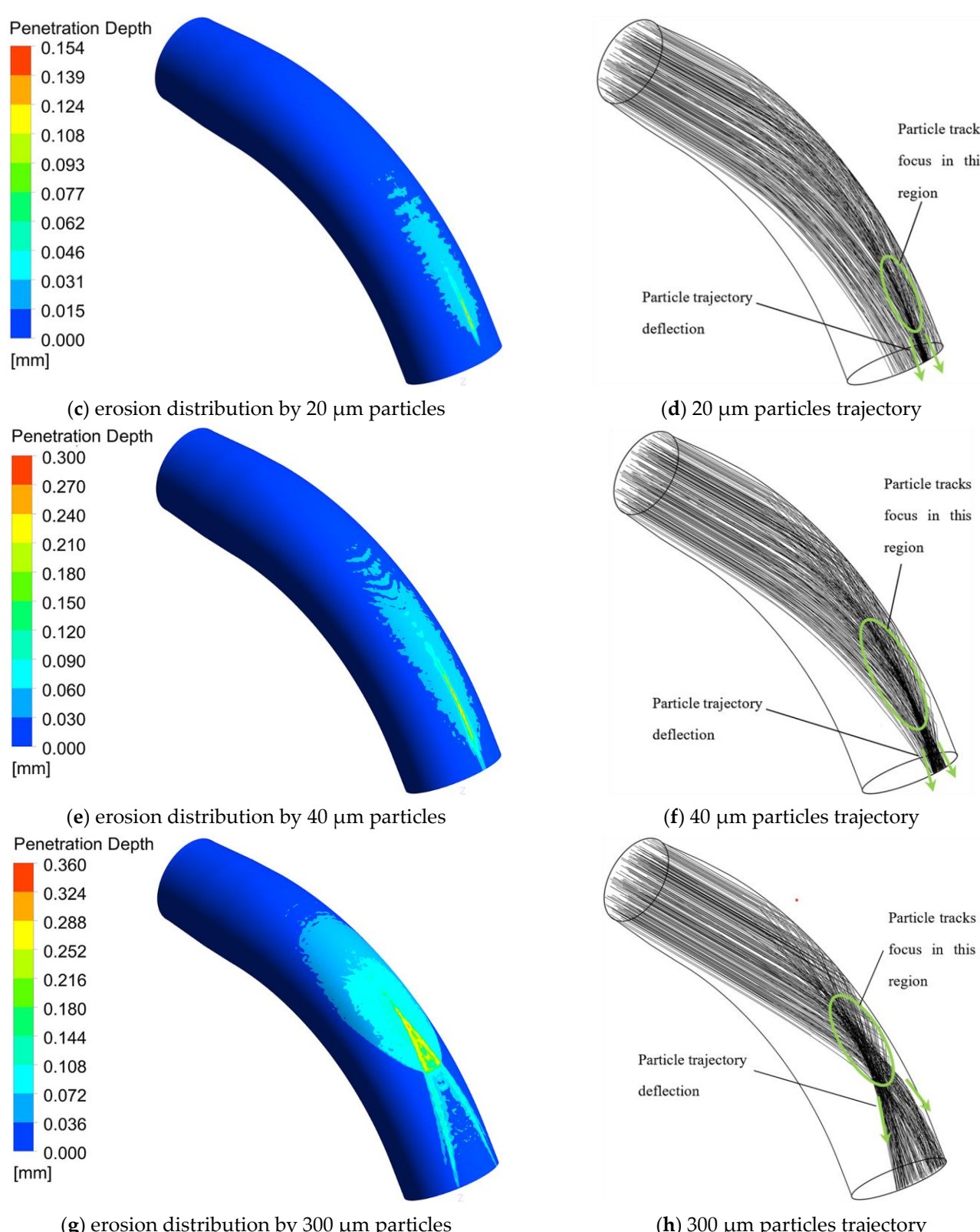

**Figure 12.** Particle erosion distribution and particle trajectory at different particle sizes.

Figure 13 shows the results of the maximum penetration depth $E_{\max}$ and its elbow curvature angle $\alpha_{PM}$, where $\alpha$ is the elbow curvature angle. It is found that when the particle size is small, the erosion depth increases with increasing particle size, then reaches

a peak at 100 μm, and then decreases slightly and gradually stabilizes, while the position of maximum erosion becomes closer to the entrance as the particle size increases.

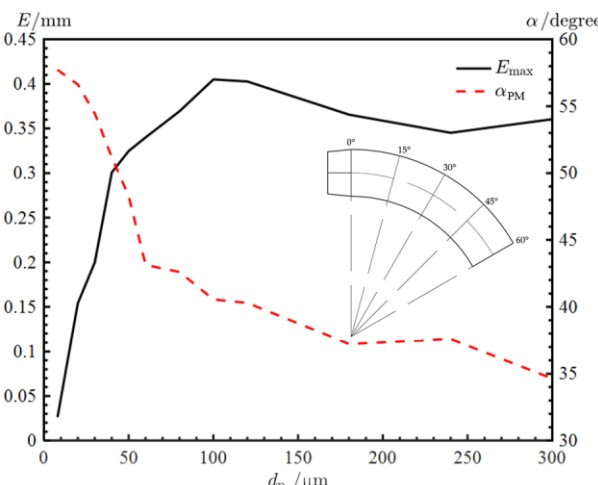

**Figure 13.** Maximum penetration depth and its elbow curvature angle at different particle sizes.

$\dot{E}_s$ and $H_T$ can also be used to assess the influence of particle size on the temperature in erosion, which is shown in Figure 14. As the particle size increases, the variation of $\dot{E}_s$ is similar to the one of $E$ in Figure 8, and the $H_T$ decreases from 3.57 to 1.63, and gradually stabilizes after 100 μm.

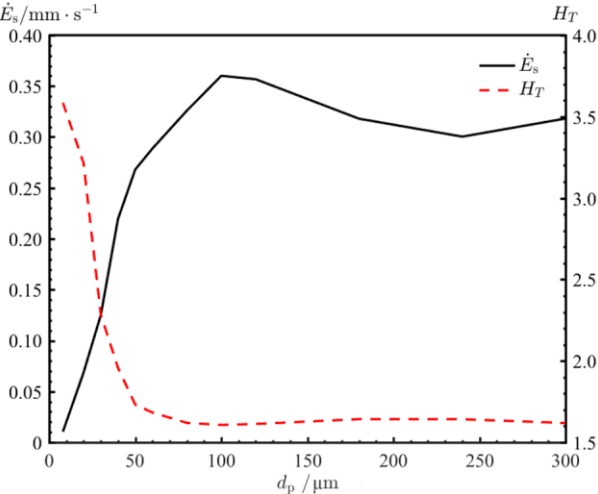

**Figure 14.** $\dot{E}_s$ and $H_T$ at different particle sizes.

To show the influence of particle size factor in the erosion model, parameter $\dot{E}_{s,d}$ is defined as the steady-state erosion rate without considering the particle size factor as follow

$$\dot{E}_{s,d} = \frac{\dot{E}_s}{f(d_p)} \tag{38}$$

To investigate the effects of particle size on the total mass of impacting particles, impact angle, and impact velocity, $\dot{m}_A$, $\overline{g}(\theta)$, and $\overline{v}$ are defined, respectively.

$\dot{m}_A$ represents particle impact mass density rate and is calculated as follow

$$\dot{m}_A = \frac{1}{A} \sum_{i(A)} \dot{m}_i \tag{39}$$

$\bar{g}(\theta)$ represents the average particle impact angle factor and is equal to the ratio of the real erosion rate to the erosion rate without considering the impact angle factor as follows

$$g\overline{(\theta)} = \frac{\dot{E}}{\frac{1}{A\rho_{elb}}\sum_{i(A)}\dot{m}_i e_{r,\theta,i}} \tag{40}$$

$$e_{r,\theta} = K\left(\frac{v}{v_r}\right)^n f(d_p)h(T_w) \tag{41}$$

where $e_{r,\theta}$ represents the erosion model without the impact angle factor.

$\bar{v}$ represents average particle impact velocity and is calculated as follow

$$\bar{v} = v_r\left(\frac{\dot{E}}{\frac{1}{A\rho_{elb}}\sum_{i(A)}\dot{m}_i e_{r,v,i}}\right)^{\frac{1}{n}} \tag{42}$$

$$e_{r,v} = Kg(\theta)f(d_p)h(T_w) \tag{43}$$

where $e_{r,v}$ represents the erosion model without the impact velocity factor.

The calculation results of $\dot{E}_{s,d}$, $\dot{m}_A$, $\bar{v}$, and $g\overline{(\theta)}$ for different particle sizes are shown in Figure 15. From Figure 15a, as the particle size increases, the trend of $\dot{E}_{s,d}$ is similar to the trend of $\dot{E}_s$, but the peak is around 50 μm. The $\dot{m}_A$ decreases rapidly when the particle diameter is small, then remains stable when the particle diameter exceeds 100 μm. This phenomenon indicates that small particles are more likely to converge at the wall surface, resulting in a larger $H_T$. This is because small particles are subjected to greater acceleration by the drag force, which makes them more prone to be blown towards the wall by the gas phase.

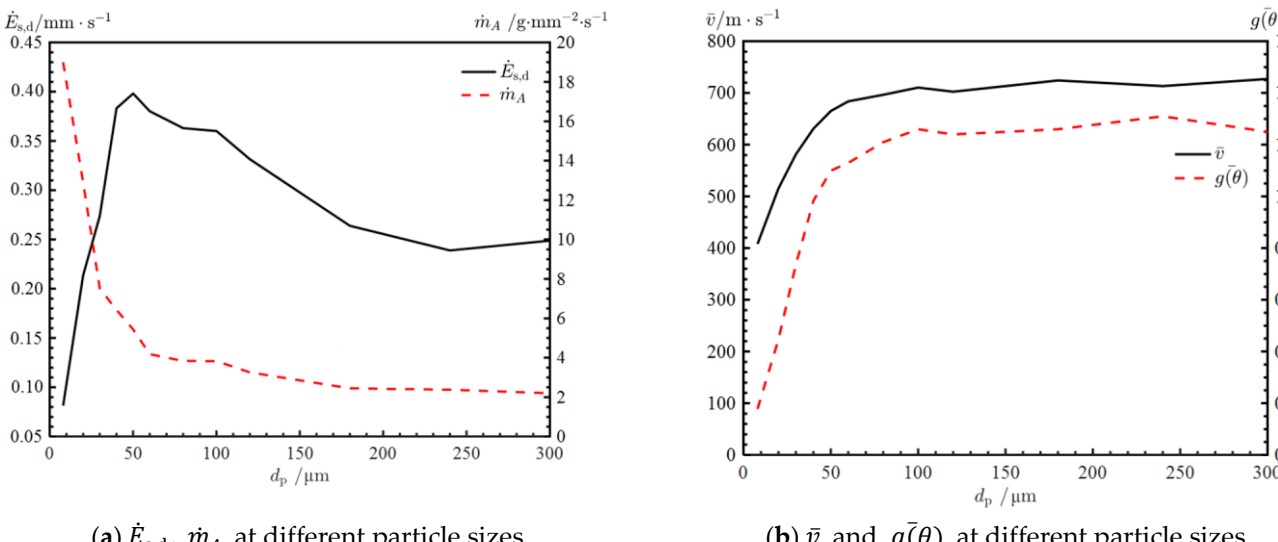

(a) $\dot{E}_{s,d}$, $\dot{m}_A$ at different particle sizes

(b) $\bar{v}$ and $g\overline{(\theta)}$ at different particle sizes

**Figure 15.** $\dot{E}_{s,d}$, $\dot{m}_A$, $\bar{v}$ and $g\overline{(\theta)}$ at different particle sizes.

According to Figure 15b, as the particle size increases, the $\bar{v}$ and $g\overline{(\theta)}$ increase, which means that the impact angle and impact velocity increase with increasing particle size.

### 5.3. Influence of Inlet Condition

The total energy of a gas-steam ejection power system can be adjusted within a certain range for launching missiles at different depths underwater, which could change the temperature and pressure of the elbow pipe's inlet. To investigate this change, this paper calculates the erosion for three inlet conditions with different particle sizes. The total

temperature and total pressure for case 1, case 2, and case 3 are 1273 K and 3.5 MPa, 1373 K and 3.75 Mpa, and 1473 K and 4 Mpa, respectively.

Figure 16 shows the maximum erosion depth and location for different grain sizes under the three conditions. The maximum erosion depth decreases along with the total temperature and pressure, but the position remains almost unchanged, indicating that temperature and pressure have little effect on particle trajectory as they decrease.

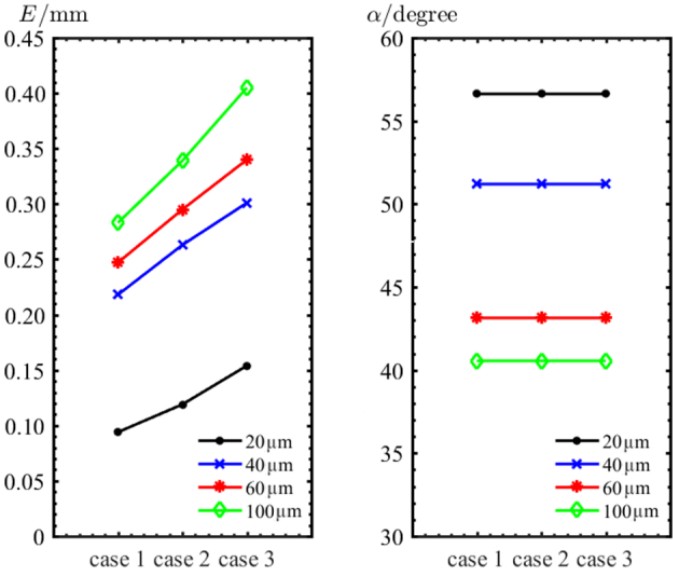

**Figure 16.** Maximum erosion depth and location of erosion for different particle sizes under three cases.

To investigate the reason for the decrease in the erosion depth, $\dot{E}_s$ and $H_T$ were calculated for different inlet conditions, and the results are shown in Figure 17. As can be seen in Figure 17, both $\dot{E}_s$ and $H_T$ will decrease as total temperature and total pressure at the inlet decrease. The decrease in $H_T$ is attributed to the decrease in wall temperature, which is caused by the decrease of heat transfer to the wall of the structure after the decrease in total temperature.

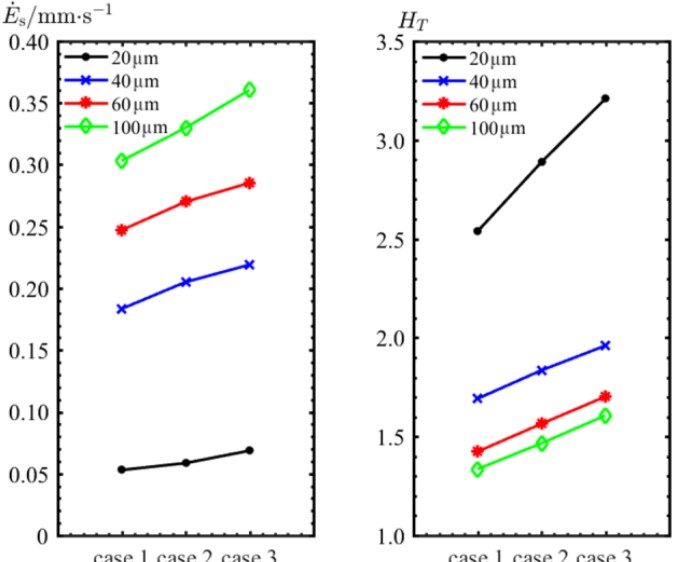

**Figure 17.** $\dot{E}_s$ and $H_T$ for different particle sizes under three cases.

Due to the small effect on the particle trajectory, it is conjectured that the decrease in $\dot{E}_s$ is caused by the particle impact velocity. The steady-state erosion without velocity factor is defined as follows

$$\dot{E}_{s,\overline{v}} = \dot{E}_s \Big/ \left( \frac{\overline{v}}{v_{\text{ref}}} \right)^n \tag{44}$$

The calculation results of the average velocity and $\dot{E}_{s,\overline{v}}$ are shown in Figure 18. As the total inlet temperature and pressure decrease, the average particle impact velocity decreases, whereas $\dot{E}_{s,\overline{v}}$ is almost constant, indicating that the particle impact velocity decrease is primarily responsible for the decrease in $\dot{E}_s$.

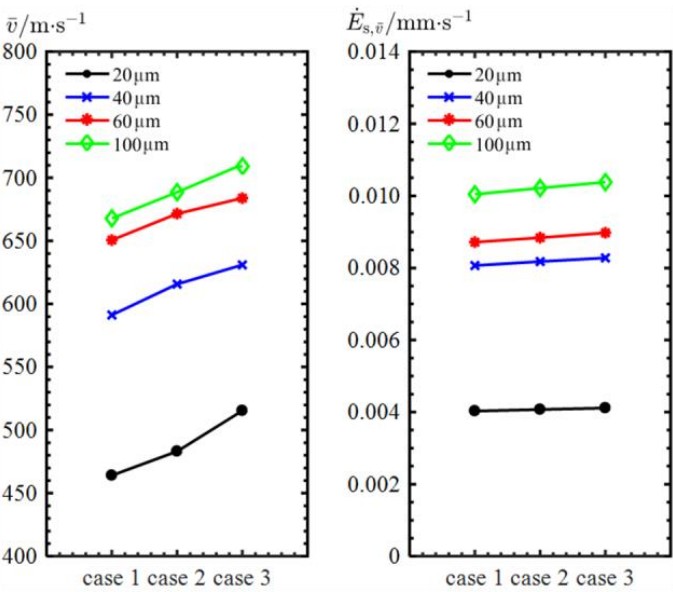

**Figure 18.** $\overline{v}$ and $\dot{E}_{s,\overline{v}}$ for different particle sizes under three cases.

### 5.4. Influence of Mass Flow Rate of Particle

To study the influence of particle mass flow rate on the erosion of the elbow pipe, the erosions were calculated for 20-, 30-, 40-, and 200-μm diameter particles at mass flow rates of 5, 10, 15, 25, and 30 kg·s$^{-1}$, respectively. The results are shown in Figure 19, where the $\dot{E}_{s,m}$ represents the steady-state erosion rate per unit mass flow rate, that is

$$\dot{E}_{s,m} = \frac{\dot{E}_s}{\dot{m}_p} \tag{45}$$

From Figure 19a, $\dot{E}_{s,m}$ indicates the particle erosion per unit mass flow rate, and it can be found from the curve that $E_{s,m}$ is not a constant value and changes with the increase of mass flow rate, and this is affected by the particle size. For 20 μm and 30 μm particles, $\dot{E}_{s,m}$ first increases rapidly as the mass flow rate increases but then decreases as the mass flow rate increases further. Moreover, for 40-μm as well as 200-μm particle size, $\dot{E}_{s,m}$ keeps decreasing gradually as the mass flow rate increases.

From Figure 19b, it can be obtained that due to the higher temperature of solid particles relative to the gas phase, the gas-particle flow heat transfer to the inner wall of the elbow pipe intensifies with increasing mass flow rate, and $H_T$ keeps increasing. At the same time, because this heat transfer is more intense under small particle size, $H_T$ changes more obviously with mass flow rate when the particle size is small, while $H_T$ almost does not change with mass flow rate when the particle size increases to 200 μm.

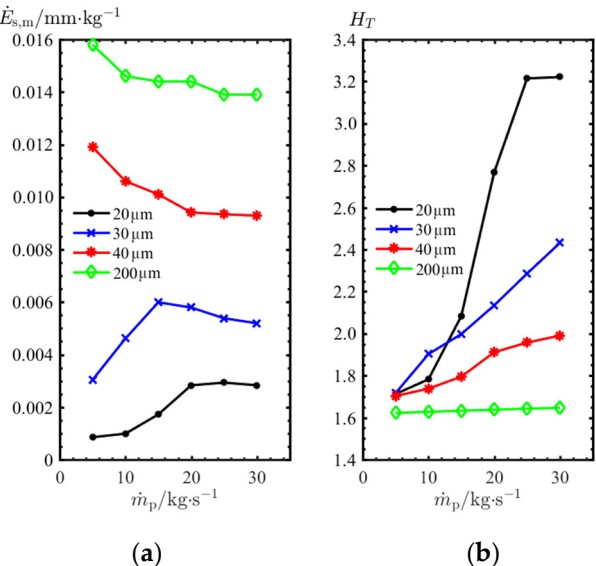

(**a**)                     (**b**)

**Figure 19.** $\dot{E}_{s,m}$ and $H_T$ with different particle sizes at different mass flow rates.

Figure 20 shows the penetration depth diffusion under 20 μm diameter particle and 5 kg·s$^{-1}$ mass flow rate. Comparing Figure 12c, it can be found that the erosion distribution is not as concentrated as the condition under the 25 kg·s$^{-1}$ mass flow rate. Figure 21 shows the penetration depth diffusion under the 40-μm diameter particle and 5 kg·s$^{-1}$ mass flow rate. Hence, comparing with Figure 12e, the angle of the particle deflection is reduced.

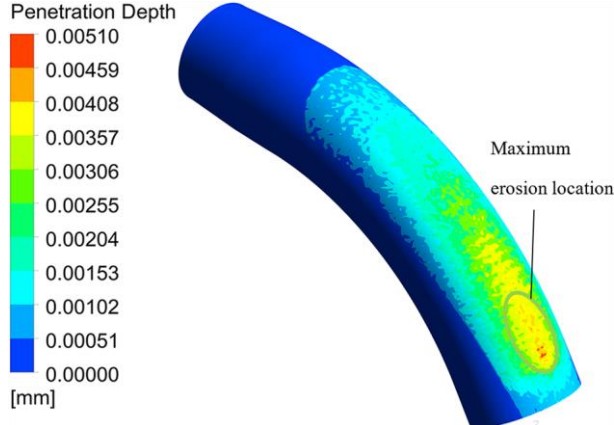

**Figure 20.** Erosion distribution in the condition of 20 μm particle, 5 kg·s$^{-1}$ mass flow rates.

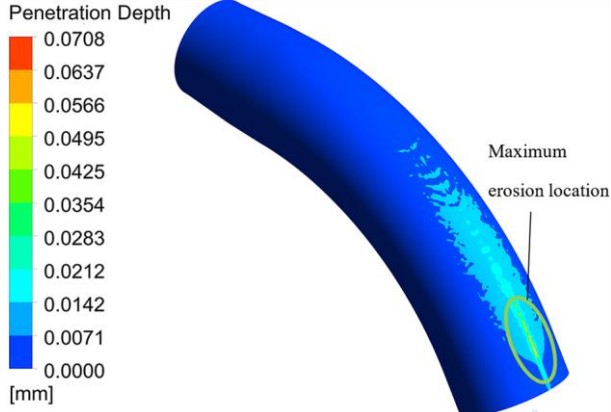

**Figure 21.** Erosion distribution in the condition of 40 μm particle, 5 kg·s$^{-1}$ mass flow rates.

Figures 19–21 illustrate that the effect of particles on gas flow is not negligible, and the changes in gas phase flow affect the trajectory of particles in turn and increase or reduce the erosion of particles on the inner wall of the elbow pipe.

To study the effect of particles on the gas phase, the pure gas flow and the gas-particle flow with different particle sizes and mass flow rates were compared at the two elbow cross-sections, as shown in Figure 22.

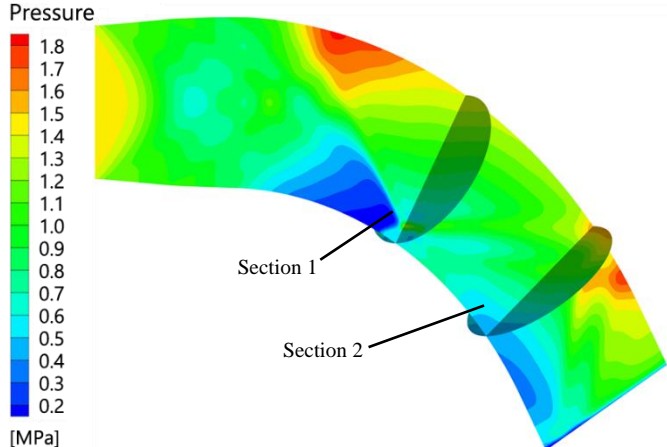

**Figure 22.** Two cross-sectional locations.

Figure 23 shows the velocity flow lines at the two cross sections under pure gas phase flow. At section 1, the flow appears separated in the upper half of the region, which leads to a flow from the symmetrical side to both sides at the outer inner wall position. At section 2, the flow appears as a Dean vortex [42,43] and the flow further flows from the symmetrical side to both sides.

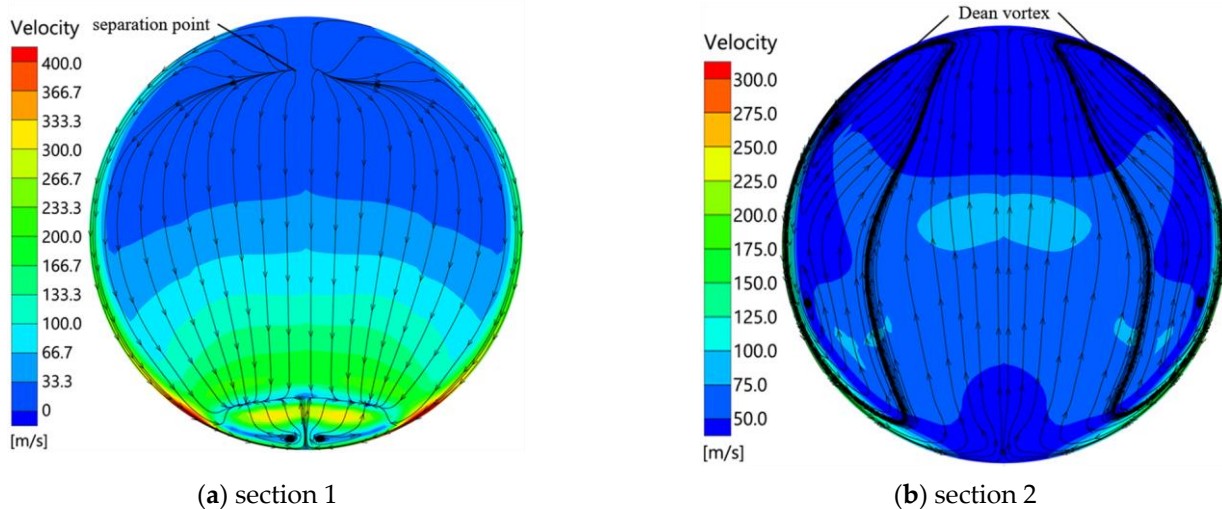

(**a**) section 1                          (**b**) section 2

**Figure 23.** Velocity flow lines on two sections in the absence of particles.

Figure 24 shows the velocity flow lines of the gas-particle flow at two sections for 20 μm diameter particles with 25 kg/s particle mass flow rate. At section 1, the airflow separation no longer appears in the upper half of the region, but instead, two vortices appear on both sides, and the two vortices will cause the airflow at the top position of the cross-section to flow from both sides to the symmetry plane. At section 2, the Dean vortex appears but is squeezed into the lower half of the section by the particle flow-induced vortex, which moves to the top of the section, keeping gas flowing to the symmetry plane. This flow increases the deflection angle of the particles towards the other side of the elbow

pipe, which increases the particle impact mass density and erosion at the symmetrical positions for the 20-μm diameter particle flow.

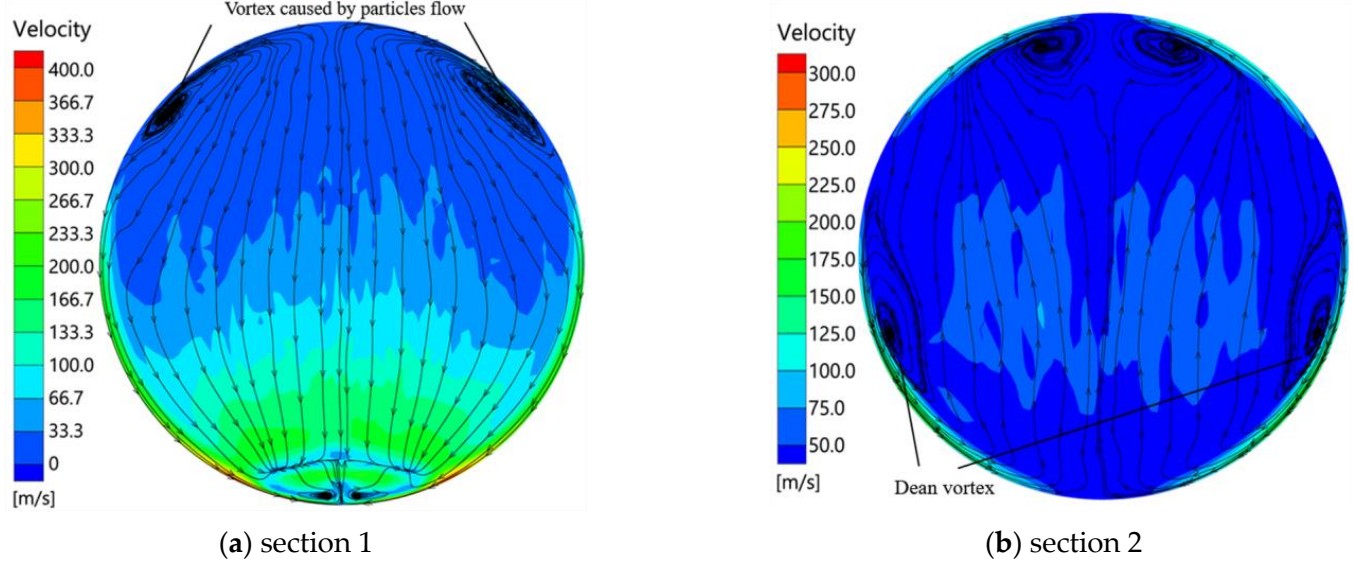

(**a**) section 1             (**b**) section 2

**Figure 24.** Velocity flow lines on two sections of gas-particle flow.

Figure 25 shows the velocity flow lines of the 20-μm particle flow at section 1 for 15 kg/s and 10 kg/s particle mass flow rates, respectively. It can be found that as the particle mass flow rate decreases, the vortex flow caused by the particles decreases or even disappears.

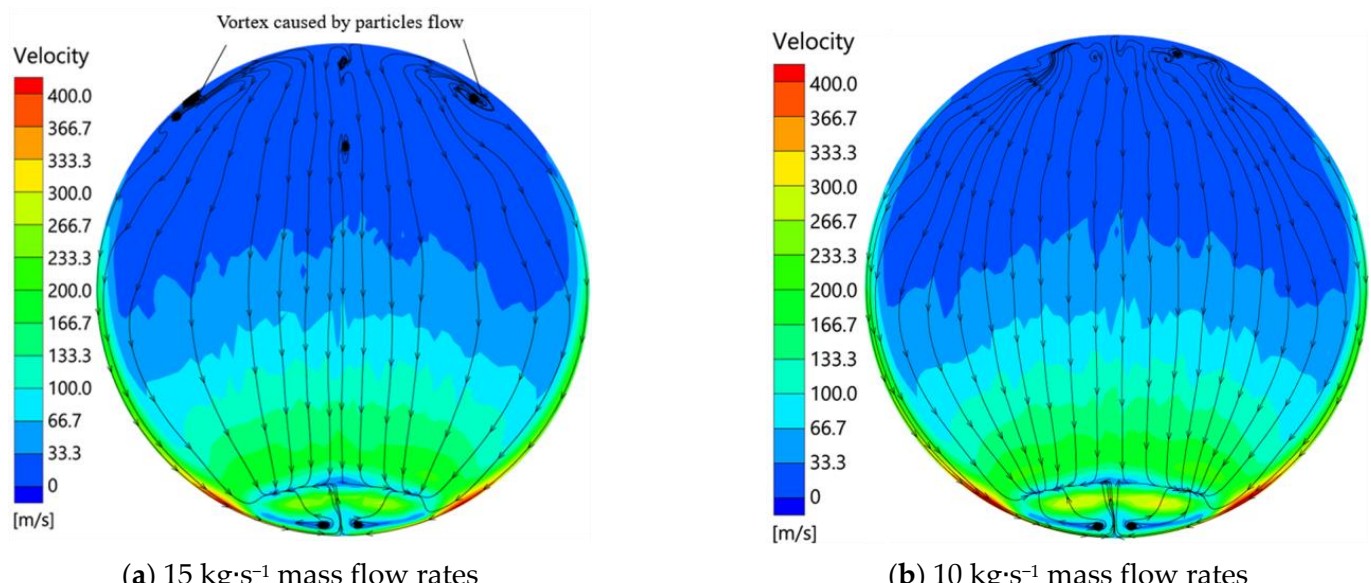

(**a**) 15 kg·s$^{-1}$ mass flow rates           (**b**) 10 kg·s$^{-1}$ mass flow rates

**Figure 25.** Velocity flow lines at 20 μm particle with different mass flow rates on section 1.

Figure 26 shows the velocity flow lines at cross-section 1 of 50 μm and 200 μm diameter particle flow for 25 kg·s$^{-1}$ particle mass flow rate. The vortex flow caused by particles decreases with decreasing particle size, and the flow separation point may appear again, causing the flow at the top of the cross section to flow from the middle to the sides.

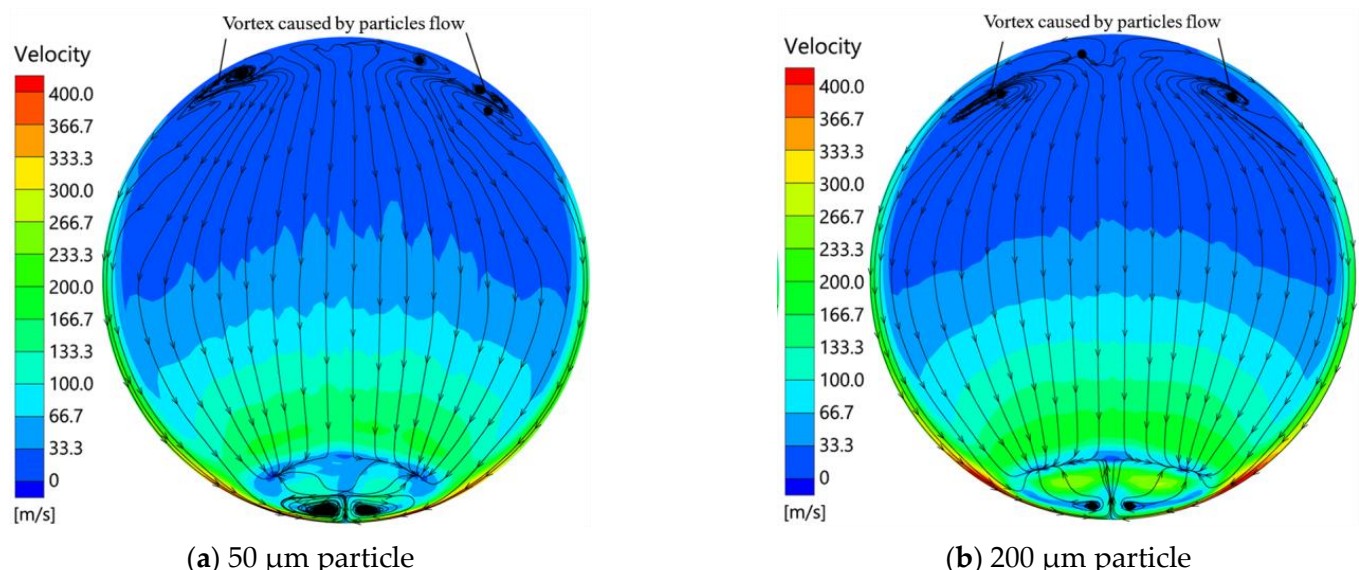

(**a**) 50 μm particle    (**b**) 200 μm particle

**Figure 26.** Velocity flow lines at different particle sizes for 25 kg·s$^{-1}$ mass flow rates.

Depending on the particle size, there will be a vortex in the gas phase. The vortex will weaken the original elbow pipe symmetry on both sides of the flow. It may even make it so the flow from each side is in symmetry, but this creates a different erosion effect. The deflection angle after particle collision is small for small particles. When the vortex intensity caused by the particle phase is low, the vortex will promote the convergence of particles at the symmetry surface and thus intensify the erosion. However, as the vortex intensity increases, the deflection angle after particle collision increases to a certain degree. This is not conducive to the convergence of particles at the symmetry surface, thus slowing down the erosion. In the case of large particles, the post-collision deflection angle itself is larger, and even if the vortex strength is lower, it is not conducive to the convergence of particles at the symmetry plane, which leads to the slowdown of erosion. The critical value of particle size for this difference is between 30 and 40 μm. As the particle size increases, on the one hand, the intensity of the vortex caused by the particles decreases. On the other hand, the trailing force of the gas phase on the particles decreases, and the influence of the gas on the particles decreases. Finally, the erosion and the mass flow rate are nearly proportional.

## 6. Conclusions

The research in this paper has led to the following conclusions:

1. By combining a discrete phase model with a flow-thermal coupling model and introducing wall temperature parameters into the erosion model, a modelling approach is developed to solve the problem of particle erosion in high temperature gas-particle flow, where the coupling of heat transfer and erosion has not been studied.

2. As particle diameter increase from 8 μm to 300 μm, the maximum erosion position moves to the inlet, and the erosion increases first and then decreases, with the peak value 0.418 mm around 100 μm particle diameter.

3. As the total inlet temperature and pressure decrease from 1473 K and 4 MPa to 1273 K and 3.5 MPa, the particle trajectory changes slightly. The location of maximum erosion depth remains unchanged. Meanwhile, the maximum erosion decreases by 20–30% while the particle velocity decreases.

4. On the outside of the elbow pipe, particles generate two counter-rotating vortices whose intensity is affected by particle size and mass flow rate. Influenced by the two vortices, the erosion per unit mass flow rate changes with the particle mass flow rate, which can even be up to double when the particle diameter is below 40 μm.

It should be noted that the assumptions made in the physical model introduce some errors, and the approach lacks direct experimental verification of high temperature gas-

particle flow heat transfer erosion, which needs to be further investigated in the future, as well as the erosion model of the elbow pipe material at high temperatures.

**Author Contributions:** Conceptualization, Q.C.; Data curation, Q.C.; Formal analysis, Q.C.; Methodology, Q.C.; Project administration, G.L.; Resources, G.L.; Supervision, G.L.; Writing—original draft, Q.C.; Writing—review & editing, G.L. All authors have read and agreed to the published version of the manuscript.

**Funding:** This research received no direct external funding.

**Institutional Review Board Statement:** Not applicable.

**Informed Consent Statement:** Not applicable.

**Data Availability Statement:** Not applicable.

**Conflicts of Interest:** The authors declare no conflict of interest.

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
