# Peer review of "Numerical Investigation on Particle Erosion Characteristics of the Elbow Pipe in Gas-Steam Ejection Power System"

_aerospace, doi:10.3390/aerospace9110635_

Round 1
Reviewer 1 Report
The article presents a numerical study of erosion by Al2O3 particles in elbow pipes used in combustion applications. The numerical method involves simulating fluid flow, heat transfer and particle model using the Eulerian-Lagrangian approach. A correlation is developed showing the influence of particle size, temperature, and pressure on particle erosion is presented using the numerical simulations. The article lacks some of the key explanations regarding the numerical method used and the following points are recommended to be addressed by the author prior to be accepted for publication.
1) Which tool is used to simulate the coupled fluid flow-heat transfer- particle model physics equations?
2) Briefly describe the modeling code or tool used in the article. Please include references or studies where the modeling tool or code is validated.
3) The article lacks experimental validation of the approach. Please include experimental validation results.
4) The conclusion of the article lists the overview of the study point by point. It lacks how the modeling approach adds value to the scientific community, the advantages, and disadvantages of the adopted modeling approach, improvements needed for the approach, and future work.
5) The introduction section of the article requires more literature reviews of existing methods related to particle erosion. How does the study add value to the scientific community? What are the advantages of the current study? How it could be extended for future studies?
Reviewer 2 Report
The manuscript needs extensive editing of English to improve quality and soundness of content. Please see the annotated manuscripts for detailed comments.

Round 2
Reviewer 1 Report
The author has significantly improved the manuscript and the article can be considered for publication.
Reviewer 2 Report
No comments. After the revision, the manuscript is suitable for publication